# Benefits and Prerequisites Associated with the Adoption of Oral 3D-Printed Medicines for Pediatric Patients: A Focus Group Study among Healthcare Professionals

**DOI:** 10.3390/pharmaceutics12030229

**Published:** 2020-03-05

**Authors:** Maria Rautamo, Kirsi Kvarnström, Mia Sivén, Marja Airaksinen, Pekka Lahdenne, Niklas Sandler

**Affiliations:** 1HUS Pharmacy, HUS Helsinki University Hospital, Stenbäckinkatu 9 B, 00290 Helsinki, Finland; kirsi.kvarnstrom@hus.fi; 2Faculty of Pharmacy, University of Helsinki, Viikinkaari 5 E, 00014 Helsinki, Finland; mia.siven@helsinki.fi (M.S.); marja.airaksinen@helsinki.fi (M.A.); 3Pharmaceutical Sciences Laboratory, Åbo Akademi University, Tykistökatu 6 A, 20520 Turku, Finland; niklas.o.sandler@gmail.com; 4Department of Children and Adolescents, HUS Helsinki University Hospital, Stenbäckinkatu 9, 00290 Helsinki, Finland; pekka.lahdenne@hus.fi

**Keywords:** children, pediatrics, 3D printing, 3D-printed medicines, personalized medication, healthcare professionals, hospital pharmacy, focus group study

## Abstract

The utilization of three-dimensional (3D) printing technologies as innovative manufacturing methods for drug products has recently gained growing interest. From a technological viewpoint, proof-of-concept on the performance of different printing methods already exist, followed by visions about future applications in hospital or community pharmacies. The main objective of this study was to investigate the perceptions of healthcare professionals in a tertiary university hospital about oral 3D-printed medicines for pediatric patients by means of focus group discussions. In general, the healthcare professionals considered many positive aspects and opportunities in 3D printing of pharmaceuticals. A precise dose as well as personalized doses and dosage forms were some of the advantages mentioned by the participants. Especially in cases of polypharmacy, incorporating several drug substances into one product to produce a polypill, personalized regarding both the combination of drug substances and the doses, would benefit drug treatments of several medical conditions and would improve adherence to medications. In addition to the positive aspects, concerns and prerequisites for the adoption of 3D printing technologies at hospital settings were also expressed. These perspectives are suggested by the authors to be focus points for future research on personalized 3D-printed drug products.

## 1. Introduction

The utilization of three-dimensional (3D) printing technologies as innovative manufacturing methods for drug products has recently gained growing interest among academia and pharmaceutical companies, with the US Food and Drug Administration (FDA) licensing the first 3D-printed medicine, Spritam^®^ by Aprecia Pharmaceuticals, in 2015. 3D printing technology aims at constructing three-dimensional objects by depositing layers of materials on top of each other, based on computer-aided design, by means of various printing techniques [1]. Studies on the development of different printing technologies in drug manufacturing of oral dosage forms are comprehensively presented in the literature. Semi-solid extrusion (SSE) has, for instance, been used to print immediate release tablets of levetiracetam [2] and paracetamol [3], chewable solid dosage forms containing isoleucine [4], as well as orodispersible films containing warfarin [5,6]. Other printing technologies used for the production of oral films include inkjet printing, also referred to as 2D printing [6,7,8,9,10]; flexographic printing [11]; fused deposition modeling (FDM) [12]; and hot melt ram extrusion [13]. Different shapes of solid dosage forms have been printed using various technologies, such as stereolithography (SLA) [14] and FDM [15]; for example, a chewable soft dosage form in the shape of a Lego brick with gelatin-based matrix and extruded drug paste inside the matrix was fabricated using embedded 3D printing [16]. Different printing technologies have also been investigated for the manufacture of drug products with sustained release profiles [17,18,19] as well as polypills incorporating more than one drug substance [14,17,20,21]. A more comprehensive description of different printing techniques used for the fabrication of drug products is presented in some recent review articles [22,23,24,25,26].

Three-dimensional printing has been presented as an accurate manufacturing method for small and personalized doses for pediatric patients, and it is thought to be suitable for on-demand manufacturing [27]. A vision of community or hospital pharmacies as compounding sites for patient-specific 3D-printed drug products has furthermore been proposed in literature [7,22,27,28,29]. Araújo et al. introduced the idea of cooperation between the pharmaceutical industry and compounding pharmacies [22]. The pharmaceutical company would provide the compounding pharmacy with raw materials and guidance on printing, whereas the compounding pharmacy would compound personalized medicines for patients with different needs based on a prescription by a physician. To the best of the author’s knowledge, there is, however, published evidence only on one printed dosage form compounded in a hospital pharmacy that has been used for children in a clinical trial [4].

As described previously in the introduction, there are numerous published studies in academic research on 3D printing of pharmaceuticals focusing on the technological performance of different printing methods. In addition, many papers present visions about future applications in the hospital environment. However, there is lack of knowledge on the perceptions of healthcare professionals on 3D-printed medicines. The adoption of printing technologies in hospital settings depend on evidence from the academic studies on technological aspects as well as on end-user acceptance of 3D-printed drug products, end-users including both healthcare providers and children themselves. Therefore, the main objective of this study was to investigate the perceptions of healthcare professionals in a tertiary university hospital about oral 3D-printed medicines for pediatric patients. This article aims to provide information on the benefits as well as the needs for future research based on concerns and identified prerequisites.

## 2. Methods

Focus group discussions for physicians, nurses, and clinical pharmacists were organized at the Department of Children and Adolescents, HUS Helsinki University Hospital. It is a tertiary care university hospital providing specialized health care for children ranging from neonates to 15-year-olds. All areas of pediatrics are covered including pediatrics, pediatric surgery, child neurology, and child psychiatry in its catchment area in Southern Finland. In addition, the department provides care for pediatric patients from across Finland in severe cardiac problems and organ transplantation as well as other rare conditions requiring demanding tertiary care. HUS Pharmacy offers hospital pharmacy services and clinical pharmacists for the Department of Children and Adolescents.

### 2.1. Study Design and Data Collection

Focus group discussions are a qualitative research method suitable for exploring the beliefs, behaviors, or attitudes of individuals [30]. This method is especially useful for unstudied topics. A semi-structured interview guide consisting of two themes and reflecting the study aim was used to moderate the focus group discussions (Table 1). The deliberative discussion approach [31] was used since the participants were not familiar with the topic of 3D printing of drug products. At the beginning of each session, the facilitator held a brief presentation about 3D printing and showed a short video clip about inkjet printing of quick response (QR) codes on edible substrates. The video is available on https://vimeo.com/253397934. The participants were able to ask questions during the discussion. A pilot interview (*n* = 3 participants) was organized to test the functionality of the interview guide and the usefulness of the presented material. Since no alteration to the guide or the presentation was made based on the pilot, the results from the pilot interview were included in the research data. Two investigators with a background in hospital pharmacy facilitated the interviews (MR, facilitator, and KK, assistant).

It was important that the different pediatric subspecialties were comprehensively represented in the focus groups; therefore, a purposive selection of participants was used. Physicians were recruited by head physicians of the different pediatric subspecialties, nurses were recruited by a nurse director, and clinical pharmacists were recruited by the principal investigator (MR). The recruitment method was invitation by email, and all participants received written information about the study. Participation in the study was voluntary, and everyone gave their written informed consent before attending the interviews. The same healthcare professionals participated in our recent study on oral drug administration practices at hospital wards [32].

### 2.2. Qualitative Analysis

The focus group discussions were digitally audio-recorded and transcribed verbatim. The data was analyzed using inductive content analysis. First, essential quotes in the transcripts were identified and coded. All codes with a similar meaning were systematically rearranged into subcategories and categories using Microsoft Excel (Microsoft Corporation, Redmond, WA, USA). The transcripts were separately analyzed by two investigators (MR and KK). Any differences in the analysis were discussed until mutual opinion was reached.

### 2.3. Ethics

The Ethics Committee of Helsinki University Hospital granted ethical approval (HUS/3637/2017, 14 December 2017).

## 3. Results

Five focus group discussions were carried out between May and September 2018. Each focus group included three to five participants from only one profession (physicians, nurses, or pharmacists), with a total number of 19 participants (Table 2).

Four main themes emerged from the focus group discussions while coding the transcripts (Figure 1). Several subcategories were also identified. The same subjects were discussed from various perspectives and brought up in different contexts during the discussion. Medication safety, for example, was an issue that was identified as a subcategory in the following three out of four main themes: benefits of 3D-printed drugs, concerns regarding 3D printing, and prerequisites for the adoption of 3D printing at hospitals.

### 3.1. Benefits of 3D-Printed Drug Products

In general, the focus group participants thought that there are many positive aspects and opportunities in 3D printing of pharmaceuticals. The possibility to manufacture a precise, patient-specific dose was seen as a significant benefit. Furthermore, the option to receive new drug products on-demand from the hospital pharmacy if the patient’s dose is changed was considered useful. A possibility to take into account the preference of each child regarding the size and form of the drug product was brought up, especially by the nurses and pediatricians. Many ideas contributing to improving drug acceptance, such as size, color, and funny or appealing shape of the medicine were introduced in the conversations. Administration of the medicine directly into the child’s mouth and the dispersion of the formulation immediately upon contact of tongue or mucosa was considered as a favorable feature.

The imprinted QR code on the orodispersible film shown in the video was considered to improve medication safety. Furthermore, dispensing errors were thought to diminish if the medicine could be printed directly on-demand based on an electronic prescription by the physician.

Incorporating several drug substances into one product to produce a polypill was seen as a remarkable benefit for the child. Polypharmacy is a reality for many children, e.g., after organ transplantation, and the possibility to combine more than one drug substance individually into a polypill was considered to improve medication adherence. Administration of one polypill instead of several medicines was regarded advantageous also for outpatients in situations when children have to be medicated outside of their home, for instance, during school days. It is also easier to remember to take all medications when the total number of drug products that has to be administered is smaller. Possibilities of cost savings regarding working tasks and hours as well as waste costs were discussed in one of the focus groups consisting of pharmacists.

### 3.2. Concerns Regarding 3D Printing of Drug Products

The healthcare professionals expressed some concerns associated with medication safety and other factors influencing the use of medicines (Table 3). Quality aspects, including dose accuracy, quality control, stability, and shelf-life of formulations, as well as the identification of drug products at hospital wards were the main concerns regarding medication safety. The focus group participants were worried that the printed dosage form would be too big in size and, therefore, it would not be possible to administer these formulations to all pediatric patients, especially infants and children having an enteral feeding tube. A concern for the functionality of a patient-specific approach to medication supplies at hospital wards was expressed as questions regarding logistical aspects and the production and delivery times of on-demand manufactured dosage forms.

### 3.3. Prerequisites for Adoption at Hospital Settings

According to the focus group discussions with healthcare professionals, the prerequisites for the adoption of 3D-printed dosage forms in the treatment of pediatric patients at hospital wards are related to subcategories of medication safety, drug administration, and production and delivery on-demand (Figure 2). Product quality is of utmost importance; hence, the drug content and accuracy of doses should be verified. The printed pharmaceuticals should be stable at room temperature and should have as long a shelf-life as possible. There must be a suitable method for product identification to prevent medication errors. In case of printing polypills, a pharmacist should check possible clinical interactions between drug substances included in the formulation before manufacturing.

The size of the printed dosage form ought to be small enough to be suitable for children. For the purpose of dissolving or dispersing the dosage form prior to administration, it has to fit into an oral syringe. In addition, it must be possible to dissolve or disperse the formulation before administering through an enteral feeding tube, e.g., for patients in intensive care or patients connected to a breathing machine. The dissolved or dispersed medicine must not block the tubes. The medical treatments often have to begin rather quickly, and the response time for production and delivery at the hospital pharmacy have to meet these needs.

### 3.4. Suggestions for Printed Medicines

Healthcare professionals proposed various needs for printed drug products, and these suggestions reflected their personal experiences from different pediatric subspecialties. Both drug substances and medical conditions were mentioned as possible targets for drug development, where 3D-printed drug products could solve the current drug administration problems experienced at hospital wards (Table 4). Furthermore, a combination product including paracetamol and ibuprofen was considered useful for children. An oral dosage form of penicillin for children as well as a suitable slow release dosage form of methylphenidate for pediatric patients were also identified as a need, especially for outpatients. Although penicillin is available as an oral solution for pediatric use, the dose volume is high and more suitable dosage forms were requested.

## 4. Discussion

### 4.1. Main Findings

Overall, the healthcare professionals participating at the focus group discussions had a positive attitude towards 3D-printed drug products. They could expect many opportunities that 3D printing technology could bring to medical treatment of pediatric patients at hospital settings. A precise dose as well as personalized doses and dosage forms were some of the benefits mentioned by the healthcare professionals. The innovative approach of considering a child’s preferences regarding shape and color of the dosage form involved suggestions of funny shapes like a pink smiley face, a tractor, and a frog. However, the appealing looks of drug products might lead to unintentional drug intake by children. This is unlikely to happen at hospital wards, though, as drug products are stored in cabinets out of the reach for children and nurses administer the medicines to patients. The possibility to print a small-sized dosage form was believed to be a benefit, while the large size of the dosage form was a big concern. Thus, the small size of the formulation was considered a significant prerequisite for adopting printed dosage forms for pediatric patients. Previously, two different shapes of polypills, cylinder and ring shape, having relatively small sizes of 10 mm in diameter and 3 or 6 mm in height have been produced by SLA [14]. A paracetamol and ibuprofen containing chewable soft dosage form in the form of a Lego^TM^-like brick has been developed utilizing embedded 3D printing technology [16]. The size of this dosage form was 40 × 25 × 15 mm, which would however be probably too big for a child. Awad et al. have used selective laser sintering (SLS) to fabricate miniprintlets (sizes 1 mm and 2 mm) containing paracetamol and ibuprofen [17].

In our study, both a small size and appealing shape of the printed formulation were assumed to improve drug acceptance. A study on drug acceptability of orodispersible films revealed that also caregivers suggested child-friendly and appealing shapes of formulations for children [33]. Additionally, the study showed that more than half of 3–5-year-old children liked orodispersible films very much and that almost all caregivers were willing to use the dosage form. Goyanes et al. investigated the effect of flavor and color on drug acceptability of chewable printlets administered to four pediatric patients [4]. Preferences of flavor and color were individual; however, the most preferred flavors were strawberry, orange, and lemon. Drug acceptability of different shapes of 3D-printed solid oral placebo formulations has only been investigated in adults [34,35]. In future studies on 3D printing of pharmaceuticals for pediatric patients, the size of the formulation should be emphasized as well as the development of innovative and child-friendly formulations and shapes.

Previous studies indicate that manufacturing of polypills is possible by means of 3D printing [14,17,20,21], but further product development with focus on pediatric patients and relevant drug substances for pediatrics is still needed. According to the healthcare professionals participating in the present study, the availability of polypills, personalized regarding both the combination of drug substances as well as the doses, would benefit the treatment of several medical conditions and would improve adherence to medications, especially in case of polypharmacy and when drug administration during school hours is necessary. Personalized drug combinations and doses for the treatment of HIV and tuberculosis as well as for medication administered after organ transplantation were mentioned as useful additions to current treatment options for children. Paracetamol and ibuprofen are often administered as a combination for the treatment of fever and pain, and this combination was also suggested as a potential polypill by the focus group participants. Pharmaceutical evaluation of potential clinical interactions between the incorporated drug substances prior to drug manufacturing of polypills was considered essential from the medication safety perspective. Indeed, the administration of polypills instead of several oral drug products has gained interest among polypharmacy patients as well [34]. Therefore, one objective for future investigations on polypills should be the manufacture of patient-specific doses for each drug substance included as well as ways to address potential interactions.

Aspects of medication safety was considered both in relation to the benefits and concerns associated with 3D printing as well as in the prerequisites for adopting printing technology as a manufacturing technique for pediatric dosage forms in hospital pharmacies. The idea of printing drug products directly based on an electronic prescription is a clear improvement to medication safety and the current processes of prescribing and compounding. This subject has been discussed also by Araújo et al. [22], who suggested that the electronically sent prescription would be directed for production to one of many printers in the compounding pharmacy after having been verified by a pharmacist. The focus group participants were concerned about drug content and distribution in printed pediatric medicines. Therefore, dose verification prior to drug administration was considered a prerequisite. Colorimetric and spectroscopic techniques for the analysis of drug content in printed dosage forms are proposed methods for nondestructive quality control permitting real-time batch release [36,37]. For on-demand manufacturing purposes, either handheld, user-friendly analytical devices that are quick to operate or devices that are automated and integrated to the printers have to be incorporated to the production process in order to enable quick response times in hospital pharmacies.

Identification of a drug product by scanning a Quick Response (QR) code imprinted on it gained much interest among the focus group participants, and this feature was considered to improve patient safety. Edinger et al. introduced the idea of using inkjet printing technology to print a QR code onto edible substrates using colorants [9]. Since then, other investigators have used the same printing method as well to mark an orodispersible film with a QR code aiming to include information on both the patient and the drug product [6]. The information in the QR code can be modified according to the needs or specifications of the hospital. The participants in this study believed that, in case the printing method used does not enable printing of a QR code, other measures should be taken to ensure identification of the drug product and, thus, to prevent medication errors. In drug development of pediatric medicines, one must consider that the colorant as well as other excipients are safe for use in children, since investigations have shown that neonates and infants are extensively exposed to harmful excipients during oral administration of medicines [38,39,40]. Furthermore, this study shows that healthcare professionals prefer dosage forms containing excipients that enable the administration of the dosage form through enteral feeding tubes as well as after dissolving or dispersing the dosage form in liquid, e.g., water.

The costs of adopting 3D printing as new manufacturing techniques at hospital pharmacies were discussed from two different perspectives. On one hand, waste costs for, e.g., unused drug products were thought to diminish if patient-specific dosage forms would be produced on-demand. On the other hand, there were concerns that personalized dosage forms would be expensive to manufacture. The costs for manufacture of personalized doses by means of printing would, for instance, include investment and annual maintenance costs for the printer; cost of personnel operating the printer; as well as costs for raw materials, packaging materials, and disposable manufacturing equipment. The actual costs depend on the chosen printing technique and studies of cost-effectiveness would be important subjects for further research in order to evaluate which drug substances would be beneficial to produce as personalized doses. The focus group participants were also concerned about how the logistics and the medical formulary at the hospital wards would be affected by a more personalized approach to the manufacture of pediatric drugs and doses. Currently, the wards are able to borrow medicines from other wards in case of unexpected needs. Manufacturing of personalized doses or dosage forms equals drug production on-demand, which in turn generates the prerequisites of short production and delivery times from time of prescription to time of administration. There are some studies regarding 3D printing of drug products where manufacturing times are mentioned [4,6]. However, since the production time is one prerequisite for the adoption of printed drug products at hospitals, it would be essential to incorporate more information about the overall manufacturing times (including all manufacturing steps) in research articles, hence making it possible to compare the suitability of different printing methods for on-demand manufacturing.

### 4.2. Strengths and Limitations

The qualitative study method of focus group discussions was suitable for investigating this topic, of which there is no previous evidence. A piloted interview guide as well as two investigators coding and analyzing the data independently of each other ensured reliability of the study. In focus group studies, the number of participants is often relatively small. However, incorporation of several pediatric subspecialties of participants with various professional background relevant to the research question strengthened this study. The results cannot necessary be generalized since this study was conducted in only one pediatric tertiary hospital in Finland. However, in any pediatric hospital providing specialized care for demanding patients, the challenges are most probably similar. The focus groups consisted of healthcare professionals, and their perceptions about 3D-printed drug products might differ from the opinions of pediatric patients or their caregivers.

## 5. Conclusions

The healthcare professionals saw many positive aspects in 3D printing as a manufacturing method of pediatric medicines. Development of polypills would improve drug treatments and drug adherence in pediatric patients. An ideal immediate-release 3D-printed product for children would be as small as possible, would be easy to identify at wards, would be stored in room temperature, and would have adequate shelf-life. Additionally, it would dissolve quickly after administration directly into the mouth and be suitable for administration through enteral feeding tubes, after having been dissolved or dispersed in a small amount of liquid. The manufacturing method must enable production and delivery on-demand within a short timeframe. The prerequisites for the adoption of 3D printing in hospital settings were extensively covered in the focus group discussions and gave information about the necessary features of 3D-printed drug products, particularly from a medication safety point of view. These perspectives are suggested by the authors to be focus points for future research on personalized 3D printing of pharmaceuticals.

## Figures and Tables

**Figure 1 pharmaceutics-12-00229-f001:**
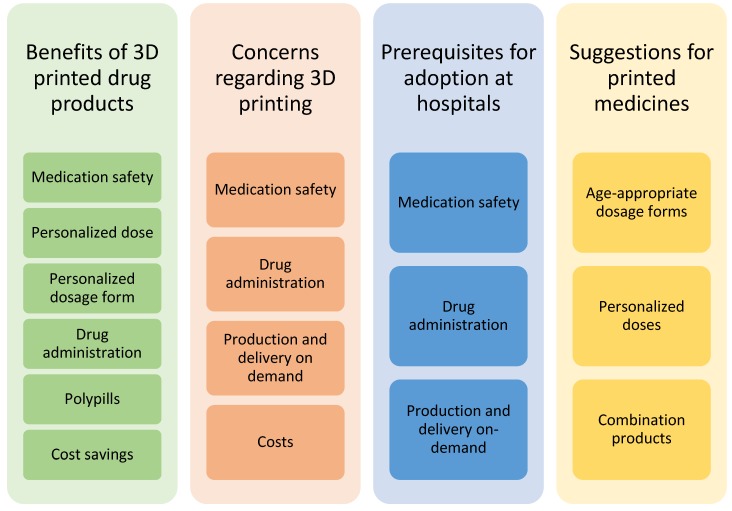
Themes of categories and subcategories identified in the focus group discussions of healthcare professionals on 3D-printed drug products.

**Figure 2 pharmaceutics-12-00229-f002:**
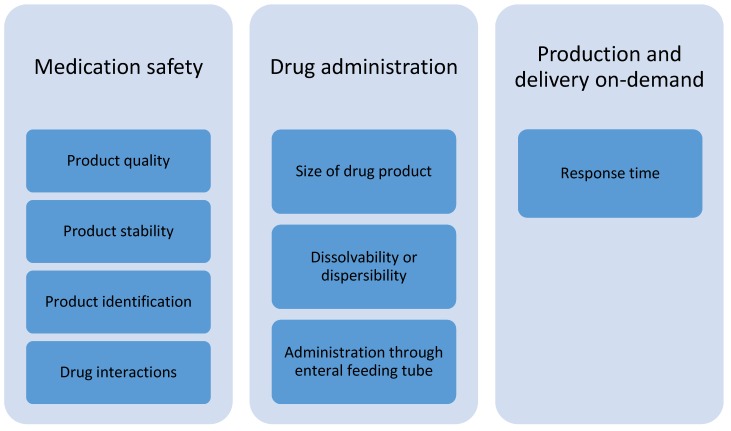
Identified subcategories of prerequisites for the adoption of 3D-printed drug products for pediatric patients at hospital wards.

**Table 1 pharmaceutics-12-00229-t001:** Interview guide for the focus group discussions about 3D-printed medicines.

Themes
3D printing as manufacturing technology for drug products thoughts about this new technology regarding benefits, concerns, and risks for which drug substances would you choose a 3D-printed drug product
Need for personalized medication for which drug substances is there a need for patient-specific doses

**Table 2 pharmaceutics-12-00229-t002:** Characteristics of the interviewed healthcare professionals (*n* = 19) in the focus groups (*n* = 5).

Variable	Physicians (n)	Nurses (n)	Pharmacists (n)
Gender			
Female	4	5	6
Male	4	0	0
Total	8	5	6
Age			
20–34	0	1	0
35–49	6	2	5
>50	2	2	1

**Table 3 pharmaceutics-12-00229-t003:** Concerns about 3D-printed drug products expressed by the healthcare professionals attending the focus group discussions.

Subcategory	Question or Comment (citations)
Medication safety	*Is the drug substance evenly distributed within the printed dosage form?*
	*Does the printing technology produce accurate doses?*
	*Does the printed dosage form perform in a desired manner and enable the optimum therapeutic effect?*
	*Is it possible to check the content of each formulation with a barcode reader?*
	*How are the different drug products identified if they look very similar?*
	*How long is the shelf-life and can the medicine be stored in room temperature?*
Drug administration	*Can the printed dosage form be dissolved or dispersed and given through a nasogastric tube?*
	*For small children, it is essential that you can print a small sized dosage form.*
	*If the printed drug products are big in size and do not dissolve/disperse in liquid, then it is not an answer to the needs of pediatric medication.*
Production and delivery on-demand	*How is the logistics at the wards affected if there are many patient-specific drug products instead of commercial packages?*
	*What is the delivery time?*
	*How fast can the hospital pharmacy react to dose changes?*
Costs	*How much more expensive are patient-specific drug products?*
	*What are the costs? Is 3D printing profitable and sensible?*
	*Does tailoring of doses give enough advantage that it is worth paying for?*

**Table 4 pharmaceutics-12-00229-t004:** Suggestions made by the focus group participants on drug substances and medical conditions where 3D-printed drug products would be beneficial.

Drug Substance or Medical Condition	Reason
Esomeprazole	Need for personalized doses of oral drug products
Ketamine	Current lack of oral drug products for pediatric patients
Midazolam	Need for better options to currently available dosage forms
Paracetamol	Need for better options to currently available dosage forms
Risperidone	Need for orodispersible dosage form
Warfarin	Need for personalized doses of oral drug products
Electrolytes	Current lack of oral drug products for pediatric patients
Strong opiates, e.g., morphine and oxycodone	Need for better options to currently available dosage forms Current lack of oral dosage forms for pediatric patients
Cancer	Need for personalized doses of oral drug products
HIV	Need for combination products and personalized doses
Organ transplantation	Need for combination products and personalized doses
Tuberculosis	Need for combination products and personalized doses

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
