# Peer review of "Benefits and Prerequisites Associated with the Adoption of Oral 3D-Printed Medicines for Pediatric Patients: A Focus Group Study among Healthcare Professionals"

_pharmaceutics, 2020, doi:10.3390/pharmaceutics12030229_

Round 1
Reviewer 1 Report
I have read the article entitled 'Benefits and prerequisites associated with the adoption of oral 3D printed medicines for pediatric patients: a focus group study among healthcare professionals' with interest, finding it to be a well thought out study with importance to the research field.
There are some minor comments and amendments that I would suggest (please see attached edited pdf file) and, whilst there are some limitations (that are acknowledged by the authors), I believe that this is worthy of publication, particularly with regards to inform future research directions.

Reviewer 2 Report
I read this paper with great interest. Whilst it is a simple study it is very relevant at this time. The topic is discussed in many forums but to date I am not aware of a real review of what it means to practitioners. It is well written and referenced and brings some clarity of thoughts to the next steps of this novel process of medicine manufacture.
Author Response
Dear Reviewer,
We are very thankful for your comments and positive feedback.
Reviewer 3 Report
The work by Rautamo et al. describes the healthcare professionals' opinion and viewpoint about the 3D printing method as manufacturing method of pediatric medicines in a hospital. Benefits, strengths and limitations of this method for the preparation of personalized dosage forms designed for children are also extensively discussed. The manuscript is well organized and can be considered of high interest to the readers. It is opinion of this reviewer that the manuscript could be accepted for publication in the current form. No revision is needed.
Author Response

(The authors gave the same response as above.)

Reviewer 4 Report
This work provides information on the perceptions of healthcare providers on 3D printed medicines. The findings are based on a very limited cohort, with no detailed guidance provided. I would not recommend it being published as an article in a scientific journal like Pharmaceutics.
Author Response
Dear Reviewer,
Thank you for your comment. Focus group discussion is a vigorous qualitative method for investigating opinions of individuals, in this case healthcare professionals. Even though the sample size appears small, it is adequate for the used qualitative method. We believe this work provides an important perspective to pediatric drug development by identifying aspects that are worth considering when developing new dosage forms or manufacturing technologies for the production of personalized doses on-demand. We also agree with other reviewers that this article gives significant information about the needs and directions for further research on 3D printing.
Reviewer 5 Report
The main objective of this study was to investigate the perceptions of healthcare professionals in a terciary university hospital about oral 3D printed medicines for pediatric patients by means of focus group discussions.
Authors describe this method as ideal immediate release 3D printed product for children would be as small as possible, easy to identify at wards, be stored in room temperature and have adequate shelf-life, and I agree but they should include some comments about the disavantages about this technic as the high cost of production or long time to obtained a minicapsule.
Can authors add some ideas about cost-effectivity process and the real possibility to use in hospitals as a individualization of dose for patients?
Author Response
Dear Reviewer,
We are very thankful for your comments and suggestions for improvement of the manuscript. We have answered your comments and marked the changes with “track changes” function in the revised manuscript.
Response: Thank you for making this suggestion of adding the aspect of cost of production and cost-effectiveness. The Discussion has been complemented with the following sentences “The costs of adopting 3D printing as new manufacturing techniques at hospital pharmacies were discussed from two different perspectives. On one hand, waste costs for e.g. unused drug products were thought to diminish if patient-specific dosage forms would be produced on-demand. On the other hand, there were concerns that personalized dosage forms would be expensive to manufacture. The costs for manufacture of personalized doses by means of printing would for instance include investment and annual maintenance costs for the printer, cost of personnel operating the printer, as well as, costs for raw materials, packaging materials and disposable manufacturing equipment. The actual costs depend on the chosen printing technique and studies of cost-effectiveness would be important subjects for further research in order to evaluate which drug substances would be beneficial to produce as personalized doses.” on L295 - 304.
Round 2
Reviewer 4 Report
The revised work provides useful suggestions on the size, shape, flavor, color, dosage, costs, and other special needs on 3D printed drugs. This can be a helpful guide for improving the acceptance of 3D printed drugs. Although the study was conducted in only one pediatric tertiary hospital in Finland, the demand can be similar. Thus, the work merits publication.